# CRISPR-Cas9 Gene Editing and Secondary Metabolite Screening Confirm *Fusarium graminearum* C16 Biosynthetic Gene Cluster Products as Decalin-Containing Diterpenoid Pyrones

**DOI:** 10.3390/jof9070695

**Published:** 2023-06-23

**Authors:** Carmen Hicks, Thomas E. Witte, Amanda Sproule, Anne Hermans, Samuel W. Shields, Ronan Colquhoun, Chris Blackman, Christopher N. Boddy, Rajagopal Subramaniam, David P. Overy

**Affiliations:** 1Ottawa Research & Development Centre, Agriculture & Agri-Food Canada, 960 Carling Ave., Ottawa, ON K1A 0C6, Canada; carmen.hicks@agr.gc.ca (C.H.); tom.witte@agr.gc.ca (T.E.W.); amanda.sproule@agr.gc.ca (A.S.); anne.hermans@agr.gc.ca (A.H.); samuel.shields@agr.gc.ca (S.W.S.); ronan.colquhoun@agr.gc.ca (R.C.); chris.blackman@agr.gc.ca (C.B.); rajagopal.subramaniam@agr.gc.ca (R.S.); 2Department of Chemistry & Biomolecular Sciences, University of Ottawa, Ottawa, ON K1N 6N5, Canada; christopher.boddy@uottawa.ca

**Keywords:** Fusarium Head Blight, virulence factor, secondary metabolism, CRISPR-Cas9, gene editing, *Fusarium graminearum*, wheat pathogen

## Abstract

*Fusarium graminearum* is a causal organism of Fusarium head blight in cereals and maize. Although a few secondary metabolites produced by *F. graminearum* are considered disease virulence factors, many molecular products of biosynthetic gene clusters expressed by *F. graminearum* during infection and their associated role in the disease are unknown. In particular, the predicted meroterpenoid products of the biosynthetic gene cluster historically designated as “C16” are likely associated with pathogenicity. Presented here are the results of CRISPR-Cas9 gene-editing experiments disrupting the polyketide synthase and terpene synthase genes associated with the C16 biosynthetic gene cluster in *F. graminearum*. Culture medium screening experiments using transformant strains were profiled by UHPLC-HRMS and targeted MS^2^ experiments to confirm the associated secondary metabolite products of the C16 biosynthetic gene cluster as the decalin-containing diterpenoid pyrones, FDDP-D and FDDP-E. Both decalin-containing diterpenoid pyrones were confirmed to be produced in wheat heads challenged with *F. graminearum* in growth chamber trials. The extent to which the *F. graminearum* C16 biosynthetic gene cluster is dispersed within the genus *Fusarium* is discussed along with a proposed role of the FDDPs as pathogen virulence factors.

## 1. Introduction

Agriculturally relevant fungal pathogens threaten global crop production, with losses occurring in all major cash crops [1]. In some cases, crop losses are further amplified postharvest due to kern.

el contamination with regulated mycotoxins, diminishing their value or prohibiting marketability, posing serious health hazards and export barriers. Although the pathogen threats themselves are known, a definitive understanding of which genes within the pathogen genome (and their associated molecular products) play a role in disease virulence often is poorly understood [2]. Virulence factors are small molecules that promote colonization and cell-to-cell transmission of a pathogen within a host. As CRISPR-Cas9 methodologies have been validated in more than 40 different species of Oomycetes and filamentous fungi (including pathogenic genera such as *Aspergillus*, *Fusarium*, and *Sclerotinia*), precision gene-editing tools have facilitated research in gene/function linkages in fungi, and in particular, show promise in the functional assignment of virulence-factor-encoding genes in plant pathosystems.

*Fusarium graminearum* is an excellent model organism for the study of virulence factors associated with Fusarium head blight (FHB), a devastating disease of cereals globally—both in terms of crop losses and contamination/infiltration of infected kernels with regulated mycotoxins. Several secondary metabolite virulence factors produced by *F. graminearum*, such as deoxynivalenol, fusaoctaxins, and gramillin [3,4,5], have been linked with the onset of disease; however, the product and function of many *F. graminearum* secondary metabolite biosynthetic gene clusters (BGCs) that are expressed during infection remain unknown [6,7]. Transcriptomic analysis of *F. graminearum* challenge in various plants (wheat, barley, and maize) led to the hypothesis that secondary metabolites associated with the BGC designated as “C16” as being “pathogenicity-related” [7,8,9,10]. Boedi et al. compared gene expression during both pathogenic (living wheat) and saprophytic growth (dead wheat) and found that genes within the *F. graminearum* C16 BGC were exclusively induced by plant signals during pathogenesis and not under saprophytic conditions [10]. Before the start of our research project, the secondary metabolite products of the *F. graminearum* C16 BGC were unknown. Expression of the C16 BGC was not previously observed in various axenic culture conditions tested [8,11], hampering efforts to characterize the molecular products of the C16 BGC through traditional purification and structural elucidation experiments. Proof of the involvement of the *F. graminearum* C16 BGC products as FHB-associated virulence factors remains to be conclusively established.

Our research group was therefore interested in using CRISPR-Cas9 gene-editing tools to elucidate the product(s) of the *F. graminearum* C16 BGC and to ascertain whether the product(s) act as a virulence factor in FHB. In *F. graminearum*, the C16 BGC is composed of genes encoding for a nonreducing polyketide synthase (PKS15; FGSG_04588), a geranylgeranyl pyrophosphate synthase (TS; FGSG_04591), a terpene cyclase (FGSG_12222), a prenyltransferase (FGSG_04593), and flavin adenine dinucleotide-dependent epoxidase (FGSG_04595)—enzymes that are proposed to be responsible for synthesizing the core chemical backbone of the molecule—in association with several additional tailoring enzyme genes (Figure 1A; Appendix A). Genome mining revealed that the core enzymes responsible for building the chemical backbone of the metabolite products of the C16 BGC are present and conserved amongst several different fungi [12]. During our research into the *F. graminearum* C16 BGC product, Tsukada et al. heterologously expressed the associated core BGC synthase genes (originating from *Arthrinium sacchari*) and various tailoring enzyme genes (from *F. graminearum*, *Colletotrichum higginsianum*, and *Metarhizium anisopliae*) in multiple iterations within *Aspergillus oryzae* [12]. Following subsequent metabolite purifications, 1D and 2D NMR and mass spectrometry experiments were used by Tsukada et al. to elucidate the molecular structures of the various BGC products [12], designated as ‘fungal decalin-containing diterpenoid pyrones’ (FDDPs), of which FDDPs -B, -D and -E were predicted to be products of the *F. graminearum* C16 BGC (Figure 1B).

Presented here are the results of gene-editing experiments disrupting the PKS15 gene and the TS gene of the C16 BGC in *F. graminearum*. CRISPR-Cas9 gene-editing protocols involving microhomology-directed repair using a hygromycin B resistance cassette were employed to disrupt selected synthase genes in the C16 BGC. Culture medium screening experiments were subsequently carried out to identify in vitro culturing conditions under which C16 BGC products were expressed. The secondary metabolite products of the *F. graminearum* C16 BGC were confirmed by UHPLC-HRMS and targeted MS^2^ experiments as FDDPs—the production of which was confirmed in planta in wheat heads upon challenge with *F. graminearum* in growth chamber trials.

## 2. Materials and Methods

### 2.1. CRISPR-Cas9 Gene Editing

*Fusarium graminearum* strain DAOMC233423 was selected for study—all media formulations used for protoplast generation and reagent formulations used in the generation of gene transformants are provided in the Appendix A. Gene-specific CRISPR-RNAs (crRNA) were designed using the Eukaryotic Pathogen CRISPR gRNA Design Tool [13]. Pairs of crRNAs were designed for each of the intended genes of interest (GOIs) with cleavage sites upstream of the predicted translation start site and near the 3′ ends of the target genes. No off-target cut sites were predicted against the *F. graminearum* PH-1 genome sequence, and care was taken to avoid disrupting adjacent genes. crRNAs were scored according to Doench et al. [14] and highly efficient crRNAs were selected. Repair template amplification primers were designed to contain 55–48 bp of homology adjacent to the gRNA-specified cleavage sites as well as 20 bp corresponding to up- and downstream regions of hygromycin resistance sequences within the pRF-HU2 vector (depicted in Appendix A).

#### 2.1.1. Generation of Protoplasts

A 200 mL volume of Littman Oxgall liquid medium was inoculated with 1 × 10^8^ *F. graminearum* macroconidia, incubated at 19 °C, and shaken at 160 rpm for 15 h, after which germling mycelium was harvested using a cell strainer (100 µm mesh size) and rinsed once with sterile water and once with 1.2 M KCl buffer. The resulting germling mycelia were transferred into a 125 mL Erlenmeyer flask containing protoplasting buffer composed of 1.2 M KCl, 500 mg Driselase, 200 mg lysing enzymes from *Aspergillus*, and 200 mg Yatalase per 20 mL and incubated at 30 °C with shaking at 80 rpm. The protoplasting buffer was prepared 45 min before use and filtered through a 0.45 µm filter prior to use. Protoplast formation was observed periodically over a total incubation time of 1 h and 15 min at which point the majority of the mycelium was found to be digested. The resulting protoplasts were diluted with an additional 20 mL of 1.2M KCl buffer, filtered through a 40 µm cell strainer, and centrifuged at 4100× *g* for 6 min at 4 °C. The pellet was resuspended and washed several times with a series of buffers (once with 20 mL of cold 1.2 M KCl buffer and twice with 20 mL of cold STC buffer (1.2 M D-sorbitol, 10 mM Tris-HCl, pH 8.0, 50 mM CaCl_2_)), each followed by a centrifugation step. The final washed pellet was resuspended in cold STC buffer and the protoplasts were quantified using a hemocytometer and diluted to a final concentration of 2 × 10^8^ protoplasts/mL in cold STC. A 7% volume of dimethyl sulfoxide (DMSO) was then added and the protoplast solution was transferred into 400 µL aliquots in sterile 2 mL Eppendorf vials and stored at −20 °C for 24 h before transfer to −80 °C.

#### 2.1.2. PCR Amplification of HDR-HygB Repair Template

A hygromycin B expression cassette (HygB) was used as a selectable marker for the Cas9-mediated gene deletion. A 1395 bp region spanning the 364 bp pTrpC and 947 bp Hygromycin sequences was amplified from the pRF-HU2 vector using primers with the addition of 35–50 bp microhomology regions specific to each intended gene deletion (Appendix A). Amplification was completed using Phusion^®^ High-Fidelity DNA polymerase (NEB, cat#M05305) and HF Phusion buffer following the manufacturer’s amplification protocol with minor modification to incorporate a touchdown start to increase primer specificity. Elongation temperatures started at 67 °C and were decreased by 1 °C per cycle for a total of 10 cycles before following the recommended 56 °C elongations. Initially, a single 50 µL reaction was completed for each of the primer sets to evaluate amplification efficacy. Once amplicon size was confirmed, additional PCR reactions (~10 reactions) were completed to obtain the desired 6–8 µg template quantity. Amplicons were then cleaned using the PureLink^®^ PCR Purification Kit (K3100-01) following the manufacturer’s protocol and a 1 µL aliquot of the cleaned template was then run on an agarose gel to confirm quality. Template quantity was determined using a Nanodrop spectrophotometer.

#### 2.1.3. In Vitro Assembly of Cas9-gRNA Ribonucleoprotein Complexes

Cas-9 ribonucleoproteins (RNPs), composed of gene-specific crRNA (Appendix A), tracrRNA, and the SpCas-9 protein, were assembled in vitro using commercially available Alt-R-CRISPR-Cas-9 components (Integrated DNA Technologies, Inc., Coralville, Iowa). Preparation of the Cas9-gRNA ribonucleoprotein complex was performed following Al Abdallah et al. [15]. In brief, two gRNA were generated corresponding to the 5′ and 3′ UTR to ensure the deletion of the entire coding sequence. Cas-9 was added to each gRNA individually, resulting in two ribonucleoprotein complexes for use in *F. graminearum* protoplast transformation.

#### 2.1.4. *Fusarium graminearum* Protoplast Transformation

For the transformation of *F. graminearum* protoplasts, 200 µL of the 2 × 10^8^ protoplasts/mL aliquots were transferred into a 2 mL tube along with 26.5 µL of RNP complex, 6–8 µg of the purified repair template, and 25 µL of a PEG (4000)-CaCl_2_ buffer (60% *w/v* PEG 4000, 50 mM CaCl_2_·H2O, 450 mM Tris-HCl, pH 7.5) followed by incubation on ice for 1 h. Afterward, an additional 1.5 mL of the PEG (4000)-CaCl_2_ was added and incubated for 20 min at room temperature. The mixture was then transferred to a 50 mL culture tube followed by the addition of 250 µL of STC buffer and 3 mL of liquid TB3 medium and incubated at a 45° angle for 18 h at 25 °C and shaking at 80 rpm. Regenerated fungal protoplasts were then combined with 20 mL of molten TB3 media containing 100 mg/L hygromycin B at volumes of 300, 500, and 1000 µL, poured into sterile Petri plates and incubated at 28 °C in the dark until the appearance of resistant colonies (5–6 d).

#### 2.1.5. Gene Deletion Confirmation and Isolation of Axenic Transformant Strains

Putative transformants for each disruption target were used to generate axenic cultures by isolation and subsequent growth of a single macroconidium. Macroconidia were generated by inoculating 10 mL CMA media with a culture explant from putative transformants before incubation at 28 °C and shaking at 160 rpm at a 45° angle for 3 d to induce conidiation. The resulting cultures were then filtered through cheesecloth to separate mycelium from conidia. Conidia were pelleted from the filtrate by centrifugation for 10 min at 4100 rpm and subjected to several washes with sterile water and centrifugation steps to remove traces of the CMC medium. The resulting conidia were then diluted to a final concentration of 1 × 10^3^ spores/mL. A 50 µL aliquot of each spore stock was then dispersed on PDA plates (150 mg/L hygromycin B) with a cell spreader and plates were incubated in the dark at 28 °C for 24 h to reconfirm retention of hygromycin resistance via germination of macroconidia. Fungal colonies (originating from a single macroconidium) were then transferred to individual PDA plates supplemented with 100 mg/L hygromycin B to allow for fungal biomass growth.

gDNA was isolated from the resulting axenic cultures using the E.Z.N.A.^®^ Plant DNA Kit following the manufacturer’s protocol, with the initial cell lysis completed using MP Biomedicals™ FastPrep-24™ and tubes containing a single ¼” ceramic bead and 2 mm zirconium oxide beads. PCR reactions were performed using amplification primers corresponding to the inserted hygromycin cassette as a positive control, and internal primers corresponding to the intended gene deletion (Appendix A). Final concentrations of 1× Titanium Taq buffer (with 3.5 mM MgCl_2_), 0.1 mM dNTPs, 0.08 µM of both forward and reverse primers, and 1× Titanium Taq polymerase (Clontech, Mountain View, California) were obtained with 2 µL of DNA template. Amplification was completed with an initial denaturation at 95 °C, followed by 35 cycles consisting of denaturation at 95 °C for 15 s, elongation at 60 °C for 15 s, and elongation at 72 °C for 30 s, with a final extension at 72 °C for 2 min. PCR products were visualized on a 1.5% SYBR Safe agarose gel at 100 V for 30 min. Strains that demonstrated amplification of the hygromycin gene, as well as a lack of amplification of their respective gene of interest, were retained for further study and cryopreserved for long-term storage.

#### 2.1.6. Southern Blot Analysis

Selected transformant strains and wild-type (WT) control were cultured in S1M medium at 28 °C with shaking at 170 rpm for 5 d and mycelium was then harvested via vacuum filtration and ground in liquid nitrogen using a mortar and pestle before gDNA isolation using an Illustra Nucleon Phytopure Genomic DNA Extraction Kit (GE Healthcare UK Limited, Buckinghamshire, UK) following the manufacturer’s protocol. All steps for DNA processing, hybridization, and Southern Blot analysis were carried out according to Gebbie [16].

### 2.2. Metabolite Profiling

#### 2.2.1. WT Profiling—Culturing

Culturing was completed on the following media: CS, CYA, CYS80, MEA, MMK2, NPN-A, NPN-P, PDA, Q6, YES, YES +IO, and ZM/2 (all media formulations are listed in the Appendix A). For the solid cultures, four Petri plates were inoculated with 2 × 10^5^ macroconidia from the *F. graminearum* WT strain and incubated at 28 °C in the dark for 10 d; two “negative” controls were prepared for each medium by inoculating with sterile water followed by incubation to ensure the effects of media composition could be back-subtracted during metabolomics data processing. After 10 d, 12 plugs (1 cm diameter) were harvested into glass scintillation vials for solvent extraction using acetonitrile. A further 12 plugs were harvested for ethyl acetate extraction with shaking at 200 rpm for 1 h. The resulting solvent extracts were decanted, dried down on a pin drier, and resuspended in 1.5 mL MeOH for UHPLC-HRMS analysis.

A 200 mL liquid seed culture for each liquid growth medium was inoculated with 1.3 × 10^6^ macroconidia in Erlenmeyer flasks with a 24 h initial shaking (160 rpm at 28 °C) before parsing out into 15 mL aliquots that were transferred into 50 mL culture tubes that were placed at a 45° angle and kept stationary for 10 days at 28 °C. Mycelium and broth were then separated and extracted separately in 15 mL of ethyl acetate (in 125 mL Erlenmeyer flasks) with shaking at 200 rpm at room temperature for 1 h. Solvent extracts were then decanted and dried down in borosilicate scintillation vials. Dried extracts were then resuspended in 1.5 mL MeOH and transferred into HPLC vials. Two pseudo-extracts or method blanks as well as two MeOH resuspension blanks were generated.

To assess for effects of gene-editing experiments, the WT, PKS15 deletion (ΔPKS15), and terpene synthase deletion (ΔTS) strains were all cultured on Q6 liquid medium. A 15 mL volume of Q6 medium was inoculated with 1 × 10^5^ macroconidia in 50 mL culture tubes with initial shaking for 24 h at 160 rpm before being kept stationary for 10 d at a 45° angle, all at 28 °C. Cultures were then harvested and extracted with ethyl acetate following previously described procedures.

#### 2.2.2. Wheat Head Infections

Susceptible spring wheat cv. Roblin Seeds were surface-sterilized for 10 min with a 30% bleach solution followed by two rounds of washing with sterile water before being transferred to a large petri dish with sterile-water-soaked Whatman filter paper and left to germinate in the dark at room temperature for 48 h. Germlings were then transplanted into 6-inch round fiber pots filled with a planting medium composed of 66% black earth (soil), 33% Pro-Mix BX M, and 1% lime for a total of 4 seeds per pot. Potted seedlings were grown under both LED and fluorescent lighting with a 16 h photoperiod and a temperature cycle of 18 °C/day and 16 °C/nights until week five where day temperatures were increased to 20 °C and night temperatures increased to 18 °C. A 20-20-20 (nitrogen, phosphorous, potassium) fertilization was applied to pots once weekly. Plants were grown to mid-anthesis (6–7 w), at which point *F. graminearum* inoculations were performed.

Pathogen inoculation protocols for greenhouse pathogenicity trials were carried out according to Desjardins et al. [17]. Postinoculation, the plants were returned to growth chambers with overhead misting for 30 s every hour for 48 h to help promote fungal infection and temperatures were increased to 24 °C/daytime and 18 °C/nights. At 14 d postinoculation, wheat heads were harvested and flash-frozen at −80 °C. For each head, the single spikelet used in point inoculation was removed from the remaining head and pooled per pot (*n* = 4–6 spikelets) for a total of 4 replicates/strain. Spikelets were freeze-dried using a Labconco FreeZone 2.5 L Benchtop Freeze Dryer System before being pulverized using a Retsch MM300 Tissue Lyser by grinding for 2 min at 22 cycles/s. Each ground tissue sample was solvent-extracted twice using 1 mL of acetonitrile (with 1% formic acid) for 1 h of nutation at room temperature followed by centrifugation and solvent decantation. The resulting extracts were combined and directly used in UHPLC-HRMS profiling.

#### 2.2.3. UHPLC-HRMS Profiling

Chemical profiling was completed using a Thermo Ultimate 3000 UHPLC coupled to a Thermo LTQ Orbitrap XL HRMS and an UltiMate Corona VeoRS charged aerosol detector (Thermo Fisher Scientific Inc, Waltham, MA, USA). Chromatography was performed on a Phenomenex Kinetex 1.7 µm C_18_ column (50 × 2.1 mm, 100 Å) with a flow rate of 0.35 mL/min using a mobile phase of H2O + 0.1% formic acid (solvent A) and ACN + 0.1% formic acid (solvent B). The column was held at 5% B for 0.5 min, ramped up to 95% B over 4.5 min, and held at 95% B for 3.5 min. The mobile phase returned to 5% B over 1 min and was left to equilibrate for 3 min before the next injection. The HRMS was operated in positive electrospray ionization (ESI^+^) mode with a 100–2000 *m/z* range and a resolution of 30,000 using the following parameters: sheath gas flow 40, auxiliary gas flow 5, sweep gas flow 2, spray voltage 4.2 kV, capillary temperature 320 °C, capillary voltage 35 V, tube lens 100 V, AGC target 5E5, and maximum ion time 500 ms. A reserpine standard was injected at the beginning of each sample sequence to confirm accurate calibration of the mass spectrometer and to aid in data alignment; all samples were injected in a randomized order and MeOH blanks were injected every six samples in the sample sequence to assess for metabolite carry-over and background subtraction during metabolomics data processing.

MS^n^ fragmentation was performed on select ions in subsequent experiments. MS^2^ scans were acquired at a resolution of 15,000 using an isolation width of 2 *m/z* and HCD at 30% NCE. The automatic gain control target was set at 5E5 and the maximum injection time was 1024 ms. Putative FDDP mass feature *m/z* were matched to feature masses reported by Tsukada et al. [12] (within 5 ppm mass spectral accuracy) and to FDDP-D, -E, and -B chemical standards (kindly provided by Teigo Asai). Mirror plots were generated using the top 50 fragments by relative intensity for each of the given mass features and plotted using ggplot2 [18].

#### 2.2.4. Metabolomics Data Preprocessing

A detailed explanation of parameters used for metabolomics data preprocessing, data reduction, and statistical analysis is provided in the Appendix A. UHPLC-HRMS profiles of both broth and mycelium extracts were processed together and separately from the UHPLC-HRMS profiles of the in planta extracts. All data were compiled into a representative matrix of metabolite mass features denoted with a retention time (RT) and mass-to-charge ratio (*m/z*). Preprocessing of the UHPLC-HRMS raw data files was completed using MzMine 2.53 [19]. Exported preprocessed data matrices of mass feature peak intensities were imported into R Studio for data processing, reduction, and statistical analysis in the R environment [20]. For the media study, all mass feature values were transformed into a binary presence–absence matrix in which values > 1E4 were converted to 1, and values < 1E4 were converted to 0 (following protocols by Witte and Overy [21]). The “pheatmap” R package, was used to generate the initial heatmap with the ‘ward.D2’ applied as both the row and column clustering algorithm [22] and then applied to each of the four extract/culture-type binary consensuses in order to sort all features and samples to correspond with the media consensus format.

For data comparison of the WT and gene deletion mutants extract profiles from the Q6 media study and the in planta infection trials, the maximum detected blank value for each mass feature was back-subtracted from all sample masses in order to reduce false positives and the dataset was normalized to the total ion current by sample sums. Univariate analysis was completed using the ‘MUMA’ R package which utilizes both Welch’s t-tests and the Wilcoxon–Mann–Whitney U tests for normally and non-normally distributed mass features, respectively, based on Shapiro scores when determining *p*-values [23]. Mass feature *p*-values comparing the ΔPKS15 and WT strains were then used to generate plots against their respective retention time. To further examine the features of statistical significance, features with *p*-values < 0.05 were plotted by RT and *m/z*. Visual inspection of the .RAW datafiles to confirm metabolomics results were done using Thermo Xcalibur software with a ±5 ppm tolerance on the predicted exact mass.

### 2.3. Genome Screening for the C16 BGC in Fusarium

To explore the distribution of the C16 BGC among *Fusarium* spp., we created a database of 1311 *Fusarium* genomes downloaded from the NCBI nt database (accessed December 2022), against which we performed BLASTn searches using the *F. graminearum* PH-1 C16 BGC nucleotide sequences as queries. A table of all the *Fusarium* genomes searched in this study is provided in the Appendix A (Appendix A). The results were manually examined for completeness using Geneious v2022.2.2 in the cases where partial clusters were detected. To visualize syntenic relationships between the C16 BGCs and their associated genetic neighborhoods, representative *Fusarium* spp. sequences were annotated de novo in Geneious, using PH-1 gene calls as a guide for coding sequence prediction for relevant C16-associated genes, and then compared to existing gene models published in GenBank, where available. The predicted amino acid sequences for each gene were then compared via BLASTp and visualized using Clinker [24]. Repeat Induced Point-mutation (RIP)-affected regions were calculated using RIPper [25], with at least two adjacent 500 bp windows having a RIP Composite Index value above zero needed for regions to be considered RIP-affected.

## 3. Results

### 3.1. Characterization of Gene Transformants

To verify secondary metabolites produced by the C16 BGC in *F. graminearum*, targeted gene disruption was successfully performed on FGSG_04588 (ΔPKS15) and FGSG_04591 (ΔTS). Multiple transformants for each gene of interest were obtained through selective growth on hygromycin-containing medium (confirming the successful incorporation of the HygB repair cassette) and successful disruption of the gene of interest was confirmed by PCR (Appendix A). Southern blot analysis confirmed the precision insertion of the HygB repair template into the intended gene locus and the absence of unintended integration elsewhere in the genome (Appendix A). All transformants contained the HygB repair template cassettes of the expected size following their respective digestions, while the WT strain had an expected absence of the HygB cassette. All generated transformant strains were grown on ½ PDA (with and without hygromycin) to compare culture phenotypes as well as to demonstrate retention of the hygromycin resistance (Appendix A). After 5 days of growth, both transformant strains of both disrupted gene loci (ΔPKS15 and ΔTS) had comparable growth phenotypes to that of the WT on ½ PDA (absence of hygromycin). In the presence of hygromycin, both the ΔPKS15 strains and ΔTS strains had a reduced growth rate compared to that observed in the absence of hygromycin; however, compared to the ΔPKS15 strains, the C16 ΔTS strains in the presence of hygromycin had a reduced growth phenotype with a prominent yellow/brown pigmentation. No growth was observed when the WT was cultured on the hygromycin medium.

### 3.2. Secondary Metabolite Profiling Confirms C16 BGC Disruption

To determine a growth medium condition for which the suspected C16 BGC metabolites were produced, secondary metabolite profiles were obtained for the WT *F. graminearum* strain from 12 different media formulations with different culture conditions (solid and liquid media) and solvent extractions. Metabolomics data preprocessing refined the UHPLC-HRMS data profiles down to a consensus data matrix (four conditions aggregated per medium) consisting of 304 mass features across 12 media. Constructed heatmaps of the consensus data across all media demonstrate clear patterns in mass feature production, with some mass features being constitutively detected across all media, while other mass features exhibited unique production limited to a few or a single medium (Figure 2). With respect to the production of the expected C16 BGC products FDDP-D and FDDP-E, corresponding mass features were detected in a limited number of media formulations. Detected mass feature intensities for FDDP-D and FDDP-E were the highest in Q6 liquid cultures relative to the other profiled media, where mass features associated with both molecules were detected in Q6 broth and mycelium extracts. FDDP-D and FDDP-E mass features were not detected in any of the solid culture extracts profiled. Therefore, the Q6 medium under liquid culture conditions was selected for subsequent profiling and metabolite evaluation of the engineered *F. graminearum* C16 BGC transformants.

Univariate analysis revealed statistically significant differential expression of mass features associated with the C16 BGC between the WT strain and that of the ΔPKS15 and ΔTS transformants in the Q6 medium (Figure 3). To visualize and compare the differentially expressed mass features, all mass features were plotted by retention time against the −Log10 (*p*-value) (Figure 3A). A subset of differentially expressed mass features of statistical significance (*p* < 0.05) were observed eluting in a group in a retention-time window focused around 5.5–6.5 min (highlighted by red dots in Figure 3A). The subset of statistically significant variables was then plotted by retention time vs. observed *m/z* (Figure 3B) revealing multiple pseudomolecular ions ([M + H]^+^, [M + H-H_2_O]^+^, [M + Na]^+^, [2M + Na]^+^) that, following comparison with analytical standards, were confirmed to be associated with FDDP-D and FDDP-E. Two additional differentially expressed mass features (red dots without annotations, Figure 3B) represented intermediates in the FDDP biosynthesis pathway. Figure 3C,D present boxplots comparing the spread of mass feature peak intensities observed for the FDDP-D and FDDP-E psuedomolecular ions ([M + H]^+^, [M + H-H_2_O]^+^, and [M + Na]^+^) between WT, ΔPKS15, and ΔTS strains cultured on Q6 medium. Both FDDP-D and FDDP-E production was abolished in cultures of the ΔPKS15 transformant and reduced in the ΔTS transformant strains compared to WT cultured on Q6 medium.

Mass feature identity of the aforementioned pseudomolecular ions was confirmed by UHPLC-HRMS comparison with chemical standards of FDDP-D and -E. Follow-up MS^2^ experiments were performed to further characterize the two major FDDP mass features detected in extracts of the WT cultured on Q6 medium (FDDP-D [M + H]^+^, RT5.71 *m/z* 503.3003; FDDP-E [M + H]^+^, RT6.13 *m*/z 487.3054). In each MS^2^ spectra of the corresponding FDDP [M + H]^+^ ions, the highest intensity fragment observed corresponded to a cleavage between the pyrone and the decalin-containing diterpene moieties (Figure 4; MS^2^ fragmentation hypotheses overlaid on FDDP structures with the highest intensity *m/z* underlined). Most of the other greater intensity molecular fragment *m/z* values observed were associated with further fragmentation of the pyrone moieties; which is as expected given the pyrone’s affinity for ionization in ESI^+^ conditions.

### 3.3. In Planta Metabolomic Profiling of WT and ΔPKS15 and ΔTS Transformants

From infected wheat tissue UHPLC-HRMS secondary metabolite profiles, in planta production of the targeted BGC C16 products FDDP-D and FDDP-E were observed to be abolished in the ΔPKS15 strain and nearly abolished in the ΔTS strain compared to the WT. Untargeted metabolomics screening also revealed a differential in planta production between the WT and transformant lines of three additional [M + H]^+^ mass features (RT5.18_503.3010, RT5.34_503.3011, and RT5.55_503.3004) sharing an identical *m/z* (±1.6–0.2 Δppm) to that of the FDDP-D [M + H]^+^ ion (eluting at an RT of 5.70; Figure 5A; Appendix A). Designation of the [M + H]^+^ annotation of the three aforementioned mass features was corroborated by observed pseudomolecular ion mass features associated with adduct formation ([M + Na]^+^) and neutral loss fragments ([M + H-H_2_O]^+^).

Extracted ion chromatograms (EICs) and MS^2^ experiments from WT in planta extracts and comparison to WT Q6 mycelial extracts confirmed the mass features of interest as potential FDDP isomers (Figure 5). In Figure 5A, EIC plots of the targeted *m/z* 503.3004 [M + H]^+^ ion from WT in planta extracts (green trace) were compared to WT Q6 mycelial extracts (red trace). An EIC of *m/z* 487.3054 ([M + H]^+^ of FDDP-E; yellow trace) was also included in Figure 5A for reference and comparison purposes for the MS^2^ analysis. Based on the results of the MS^2^ experiments (and comparison with chemical standards) of the two dominant *m/z* 503.3004 peaks (RT 5.51 min and RT 5.70 min), we predict that the in planta mass feature eluting at RT 5.70 min corresponds to the FDDP-D [M + H]^+^ ion. Due to concentration constraints of the targeted molecules in the in planta extracts, only one of the additional FDDP isomers (RT5.51_*m/z* 503.3004) was present in sufficient quantity in the in planta extract for informative MS^2^ analysis compared to MS^2^ results for FDDP-E (RT6.13_*m/z* 487.3054; Figure 5B) and FDDP-D (RT5.70_*m/z* 503.3004; Figure 5C) from WT Q6 media culture extracts. Mirror plot comparisons of the unknown RT5.51_*m/z* 503.3004 mass feature with the two FDDP end products (D and E) reveal the high similarity between the pyrone fragments generated within the unknown mass feature and those observed for FDDP-E; however, neither molecule shares the same [M + H]^+^ *m/z*. Although the unknown mass feature shares an identical [M + H]^+^ *m/z* with FDDP-D, there is very little MS^2^ *m/z* mass fragment overlap between the two molecules; therefore suggesting that the unknown FDDP analog (RT5.51_*m/z* 503.3004) likely shares the pyrone moiety of FDDP-E. Full structural characterization of the FDDP analog (RT5.51_*m/z* 503.3004) will be required to confirm this hypothesis.

### 3.4. Distribution of the FDDP Cluster within the Genus Fusarium

To explore the distribution of the *F. graminearum* C16 BGC among other species within the genus *Fusarium*, we created a database from 1311 *Fusarium* genomes downloaded from the NCBI nt database and performed local Blastn searches using the *F. graminearum* (PH-1) C16 BGC nucleotide sequences as queries. Gene matches were generally above 80% nt identity with a few exceptions. Evidence for the presence of the C16 BGC or C16-like clusters (with less than the full complement of 10 biosynthetic genes identified by Tsukada et al. [12]) was detected in 240 *Fusarium* genomes spread across 10 *Fusarium* species complexes (SCs) (Figure 6). Some species complexes, including the *F. concolor* SC and the *F. redolens* SC, had 100% of associated species bearing C16-like clusters, while other species complexes had species that were either mostly C16-positive (*F. burgessii* SC, *n* = 2/3; *F. sambucinum* SC, *n* = 19/35) or else showed a sparse distribution of species with C16-like clusters (*F. buharicum* SC, *n* = 2/4; *F. fujikuroi* SC, *n* = 3/49*; F. nisikadoi* SC, *n* = 1/4; *F. oxysporum* SC, *n* = 2/4; *F. tricinctum* SC, *n* = 2/5). *Fusarium* species complexes with no evidence of C16-like clusters included: *F. aywerte* SC (*n* = 0/1), *F. camptoceras* SC (*n* = 0/1), *F. chlamydosporum* SC (*n* = 0/2), *F. heterosporum* SC (*n* = 0/1), *F. incarnatum-equiseti* SC (*n* = 0/14), *F. lateritium* SC (*n* = 0/3), and the *F. torreyae* SC (*n* = 0/3). A table summarizing the major species complexes, species, and the number of genomes with C16/C16-like BGC hits is provided in the Appendix A (Appendix A).

It is of interest to note that when a homolog of the C16 BGC was identified in a given *Fusarium* sp., the respective BGC was found in all of the strains sampled for said *Fusarium* sp. However, we note that the sampling depth for each *Fusarium* spp. included in our analysis is fairly low (for most species, only between 1–4 representative strains). *F. graminearum* was heavily sampled, with 121 strains sequenced at the time of publication; of these, 120 possessed an intact C16 cluster, and one was a partial assembly. Other notable exceptions are *F. avenaceum*, with 10/12 strains showing C16-like clusters, *F. oxysporum*, with 4/705 strains, and *F. tricinctum*, where 1/5 strains show evidence of a partial C16 cluster but are missing most ‘core’ biosynthetic genes (Appendix A). Interestingly, of the four *F. oxysporum* strains with C16-equivalent BGC hits, all are classified as *formae specialis* (f. sp.): two as *F. oxysporum* f. sp. *matthiolae,* one as *F. oxysporum* f. sp. *melongenae,* and one as *F. oxysporum* f. sp. *rapae.* Additionally, whenever present, the equivalent C16 BGC is usually ‘complete’, meaning all 10 biosynthetic genes are present. However, there are some exceptions in the form of incomplete clusters and duplicated clusters. As our intent is not to assess the quality of all assemblies included in this study, we have refrained from further analysis of incomplete or duplicated clusters.

To date, there have been relatively few complete, chromosome- or telomere-to-telomere-level *Fusarium* genome assemblies from which predictions as to the chromosomal locus of the C16 BGC orthologs could be mapped. Nevertheless, we detected the C16 cluster to be within 500 kb of the 3′ end of Chromosome 2 in *F. graminearum* (strains CS3005, FG-12, and PH-1). An equivalent C16 BGC is similarly located on chromosome 2 in the assemblies of the closely related species *F. asiaticum* (FCTC 16664), *F. pseudograminearum* (CS3096), and *F. culmorum* (FcUK99). Completed *F. avenaceum* genomes (FaLH27, FaLH03, WV21P1A) each bore the BGC at a locus 75 kb from a telomere on chromosome 8. Apart from these examples, no other equivalent C16 BGC-bearing species with a chromosome-level assembly were available at GenBank. To infer the potential evolution of the C16 BGC with regards to structural rearrangements or the addition of new genes to the cluster (i.e., tailoring enzymes, transcription factors, or transporters), we produced a synteny map comparing the C16 BGC genetic neighborhoods of *Fusarium* species relevant to plant pathology and other more distantly related ascomycete plant pathogens (Figure 7). Our analysis indicates that synteny of the C16 BGC is largely conserved within the *Fusarium* species included, except for in *Fusarium xylarioides*, which shows a large region of low GC content predicted to be affected by repeat-induced point mutation (RIP), a genomic defense mechanism against transposable elements and associated duplicated DNA. Additionally, the C16 BGC appears in differing genetic environments based on loss of synteny with neighboring genes (within 20 kb on either side of the cluster) when compared between species.

## 4. Discussion

The secondary metabolite products of the *F. graminearum* C16 BGC were first structurally characterized by Tsukada et al. by transforming and heterologously expressing the associated BGC genes in *A. oryzae* [12]. It should be noted that core synthase genes (that are conserved across several fungal genera) originated from the genome of *Arthrinium sacchari*, while all associated tailoring enzyme genes originated from *F. graminearum*. A series of meroterpenoid secondary metabolite products were structurally characterized as ‘fungal decalin-containing diterpenoid pyrones’ or FDDPs—where FDDP-D and FDDP-E were confirmed in our present study as end products of the C16 BGC using gene editing in *F. graminearum*. Both metabolites were observed to be produced by *F. graminearum* during in planta challenge in wheat heads, along with several other potential FDDP products, as determined from metabolomics analysis of representative UHPLC-HRMS profiles of infected wheat head extracts and targeted MS^2^ experiments.

In a recent research study using multi-gene deletion strains to identify secondary metabolites associated with the *F. graminearum* C16 BGC, Seidl et al. [27] discerned two metabolite products (having an observed [M + H]^+^ *m/z* of 521.3109 ± 3ppm and an [M + H]^+^ *m/z* of 503.3003 ± 3ppm) that were differentially expressed under in vitro cultivation and in planta challenge in wheat heads between a “triple mutant” strain (Δ*tri5*, Δ*pks4,13*, and Δ*kmt6*) and a *pks15* gene disruption in the triple mutant background (Δ*tri5*, Δ*pks4,13*, Δ*kmt6*, and Δ*pks15*). Scaled-up cultivation yielded insufficient material for a full structural characterization of the molecules; however, based on MS^2^ experiments of culture extracts derived using C^12^- and C^13^-labeled substrates, Seidl et al. [27] proposed the candidate names gramiketides A and B (for *m/z* 521.3109 and *m/z* 503.3003, respectively) as products of the *F. graminearum* C16 BGC. MS^2^ mass fragments of the FDDP-D [M + H]+ (RT5.70_*m/z* 503.3004) observed in our study do not match those reported by Seidl et al. [27] for gramiketide B ([M + H]^+^ *m/z* 503.3003). Rather, the MS^2^ [M + H]^+^ mass fragments reported for gramiketide B [27] closely match the MS^2^ [M + H]^+^ mass fragments of the C16 BGC unknown mass feature observed in our study as RT5.51_*m/z* 503.3004 from in planta wheat head challenge (Appendix A). A potential mass feature corresponding to Seidl et al.’s proposed gramiketide A (*m/z* 521.3109) was also detected in our wild-type in planta extracts (*m/z* Δppm < ±5), albeit at detection intensities at the limit of detection of our instrument when searching using extraction ion chromatograms—the detected amounts were insufficient to carry out MS^2^ comparison for further validation. Although tempting, caution is advised before proposing candidate metabolite names for molecules that have yet to be fully structurally characterized. Structural characterization enables dereplication efforts and prevents duplicity of names assigned to molecular analogs derived from the same BGC of a given organism—which appears to be the case in the products characterized for the *F. graminearum* C16 BGC. As a surrogate C16 BGC was heterologously expressed in *A. oryzae*, and the resulting products were structurally characterized in full by Tsukada et al. [12] two years before the proposition of the gramiketide naming scheme by Seidl et al. [27], retention of the FDDP naming convention should have been given precedence by Seidl et al. to prevent naming duplicity.

In planta transcription of the *F. graminearum* C16 BGC during the onset of FHB disease in wheat, barley, and maize [8,9,28] lead to the hypothesis that the C16 BGC biosynthetic products have a role in pathogenicity, as virulence factors facilitate pathogen ingress and cell-to-cell transmission. For example, microarray expression profiles of *F. graminearum* C16 BGC genes significantly increase after 72 h postinoculation on barley and 96 h postinoculation on wheat followed by a subsequent decrease [8]. This result was recently corroborated using LC-HRMS profiling of *F. graminearum*-infected wheat heads, where postulated C16 BGC products along with DON were observed 72 h postinoculation, with concentrations increasing over time [27]. Transcriptomics experiments have also linked the co-expression of the C16 BGC with that of the fusaoctaxin BGC (C64) in *F. graminearum*, where transcripts of both BGC genes were observed to peak 64–96 h after infection in infected barley and wheat kernels followed by a subsequent decrease [7]. Both DON and fusaoctaxins are virulence factors associated with FHB onset caused by *F. graminearum*: fusaoctaxins promote cell-to-cell transmission by ultimately inhibiting/retarding cell wall callose deposition [4], and DON production is linked with the kernel-to-kernel transmission of the pathogen through the rachis in wheat [3]. Several other *F. graminearum* secondary metabolite BGCs have also been observed to be transcribed in planta for which secondary metabolite products have yet to be linked [7]; all have the potential to be involved as virulence factors with the onset of FHB disease. The trichothecene pathway transcription factor Tri6 acts as a global regulator of secondary metabolite expression that is linked with transcriptional regulation/expression of multiple BGCs that include, but are not exclusive to, BGCs for aurofusarin, butenolide, gramillins, fusaoctaxins, and trichothecenes [29]. Virulence factor expression by *F. graminearum* upon plant penetration and infection occurs as a network of events, rather than a single factor causing disease [30]. Coregulated expression of a network of virulence factors (associated with complementary biological activities) allows the pathogen to overcome the redundancies associated with the plant immune system and allows for pathogen infiltration in a variety of different hosts.

Higginsianins are structurally similar to the FDDPs produced by *F. graminearum* and arise from a closely related BGC (sharing homologous core synthase genes but different tailoring enzyme genes) found in the producing fungus *Colletotrichum higginsianum* [12]. In particular, the terpenoid portion of higginsianin B is identical to that of FDDP-D and FDDP-E in terms of molecular structure and stereochemistry (Figure 8). *Colletotrichum higginsianum* is also a fungal pathogen, causing anthracnose disease of Brassicaceae (including *Arabidopsis thaliana*). Studies using *A. thaliana* mutant models demonstrated that higginsianin B is a virulence factor that facilitates infection. Higginsianin B is proposed to suppress/impact endogenous jasmonic acid (JA) signaling pathways in plants, through the inhibition of 26S proteasome proteolytic activity [31]. During wounding or pathogen ingress in plants, 26S proteasome activity is required for JASMONATE ZIM DOMAIN (JAZ) protein degradation, where JAZ proteins are essential for de-repressing plant defense-gene-regulated JA signaling through competitive binding to JA-responsive genes [32,33]. Essentially, higginsianin B inhibits/retards aspects of a plant’s pathogen-recognition mechanism and downstream defense signaling. In terms of structural activity relationships, the observed biological activity of higginsianin B in *A. thaliana* infection models was attributed to the hydroxyl and/or the 4-isoheptenyl moieties attached to the decalin core, compared with other higginsianin analogs (higginsianin A, C, and 13-epi-higinsianin C [31]), structure moieties and associated stereochemistry that are conserved in the *F. graminearum* FDDP-D and FDDP-E molecules (Figure 8). In each of the higginsianin isoforms (A-C), the pyrone polyketide portion of the molecule is the same and, therefore, alterations to the pyrone moiety were not directly attributed as influencing the 26S proteasome inhibition in *A. thaliana* in terms of structure–activity relationships [31]. Due to the observed structural similarities of higginsianin B and FDDP-D and -E, 26S proteasome repression by *F. graminearum* FDDPs appears likely, and could act synergistically with other *F. graminearum* virulence factors to confound the innate plant immune response during pathogen ingress and cell-to-cell transmission during disease onset in wheat, barley, and maize. 

In our study, preliminary observations made during wheat head pathogen challenge are suggestive of a role of FDDP-D and -E as a virulence factor, similar to the role of higginsianin B in *A. thaliana* challenge with *C. higginsianum* in delaying the onset of the innate plant immune response. In wheat heads, once *F. graminearum* infection spreads past the epidermal layers of the developing fruit coat (typically 72 h post-inoculation), DON production facilitates hyphal infiltration and transmission through the rachis internode [3]. As the pathogen infiltrates the vascular tissues of the rachis, bleaching of the upper portion of the wheat spike can result in hyphal blockage of vascular spaces preventing apical water and solute transport. Concurrent upward and downward dissemination of *F. graminearum* toxins through the rachis into adjacent spikelets (prior to physical colonization) is also facilitated through the vascular system through hyphal excretion into xylem vessels and phloem sieve tubes [34]. Subsequently, *F. graminearum* spreads to adjacent spikelets both upwards and downward through the rachis of the wheat head. If apical bleaching has occurred, infection spread is visually most pronounced in terms of tissue yellowing and discoloration in spikelets in a basal direction down the rachis from the point of infection. From in planta growth chamber trials, a reduced disease phenotype (a delay in onset of the disease phenotype in terms of % bleaching and infected kernels) was observed in ΔTS transformant strains compared to the WT (Appendix A). Retardation of the disease progression through the reduction in the production of the FDDPs, but not a complete arrest of pathogen dissemination, highlights the fact that multiple virulence factors play a role in FHB. Oddly, abolishment of FDDP production in the ΔPKS transformant did not produce the same effect during in planta challenge as that observed from the ΔTS transformant. This phenomenon was previously reported by Gaffoor et al. when they disrupted the PKS15 gene and observed no effect on pathogenicity [11]. In previous pathology studies done with the higginsianins, the biological activity of higginsianin B was dependent upon the terpene portion of the molecule [31]. In our study, both the TS and the prenyltransferase genes were functional in the *F. graminearum* ΔPKS transformant—leading to the hypothesis that an alternate pyrone (not originating from PKS15) might have been prenylated by the biosynthetic pathway to assemble a new functional FDDP molecule, differing in molecular weight to that of FDDP’s observed from the WT strain but with an identical terpenoid moiety. Supporting this hypothesis is the observation that a number of fungal prenyltransferases have shown acceptor substrate promiscuity [35,36,37]. Our hypotheses are preliminary and follow-up experiments will need to be performed (such as disrupting the entire C16 BGC) to confirm observations made regarding FDDP involvement as a virulence factor in FHB.

Our metabolomic analysis determined that disruption of the Δ*pks15* gene in transformant strains abolished FDDP production, and yet surprisingly, the ΔTS transformant strains maintained FDDP production compared to the WT, albeit at lower observed intensities, in the Q6 medium. However, a significant reduction in FDDP production was observed in ΔTS transformant strains from *in planta* challenge experiments (Appendix A). Continued production of FDDPs as observed in vitro following disruption of the TS gene in transformant strains implies that FDDP-relevant terpene synthase redundancy exists in *F. graminearum*. The C16 BGC terpene synthase is more accurately defined as a geranylgeranyl pyrophosphate synthase or GGPPS, belonging to an enzyme family that, in eukaryotes, synthesizes GGPP predominantly from farnesyl diphosphate and isopentenyl diphosphate. We detected multiple genes in the *F. graminearum* 233,423 genome with predicted GGPPS function (*FGSG_10097*, *FGSG_01738, FGSG_01783*) in addition to the C16 BGC GGPPS disrupted in this study (*FGSG_04591*), which is consistent with previous observations in *F. graminearum* [38]. While recent work has helped elucidate the activity of some of these orthologs [39,40], the coordination of *F. graminearum* GGPPS activity, which may be compartmentalized during secondary metabolite biosynthesis, remains uncharacterized. Nevertheless, we propose the presence of multiple GGPPS orthologs as a rationale for the maintenance of FDDP production in the ΔTS transformant. Further exploration of GGPPS ortholog expression and subcellular localization during *F. graminearum* plant infection will be of interest for future studies of *F. graminearum* virulence factors.

Within the *F. sambucinum* SC, as represented in the analysis of Crous et al. [26], only the most immediate relatives of *F. graminearum* have the C16 BGC, except for *F. subtropicale* and *F. praegraminearum.* Other, more divergent, branches within the *F. sambucinum* SC were devoid of C16 BGC, including those associated with the plant pathogens/endophytes *F. longipes*, *F. poae*, and *F. langsethiae*. Outside of the *F. sambucinum* SC, however, the FDDP cluster appears to be relatively widespread, if sometimes infrequently detected (Figure 7). The most basal branch of Fusarioid fungi with a homologous C16 BGC was the *F. buxicola* SC (syn. *Cyanonectria*), with one of two whole-genome-sequenced species (*F. cyanostomum*) returning hits for all C16 BGC genes. Apart from *F. cyanostomum,* no other Fusarioid fungus outside of the genus *Fusarium*—as defined by Crous et al. [26] and referred to as the “F3” lineage designation in Geiser et al. [41]—had C16 BGC homologs; however, many described taxa have yet to be whole-genome-sequenced, making the picture far from complete. Given the diversity of species that harbor homologous C16 BGCs, we suggest that whatever role FDDPs may play in the context of *Fusarium* growth in planta, that role is likely non-host-specific. Inhibition of the jasmonic acid signaling pathway, an essential component of innate plant immunity, is a reasonable fit for this predicted role. The prevalence of plant pathogens among those *Fusaria* that we have shown to be capable of FDDP or FDDP-like secondary metabolite production underscores the need for further research into the impact of this secondary metabolite class in *Fusarium* pathogenesis.

## Figures and Tables

**Figure 1 jof-09-00695-f001:**
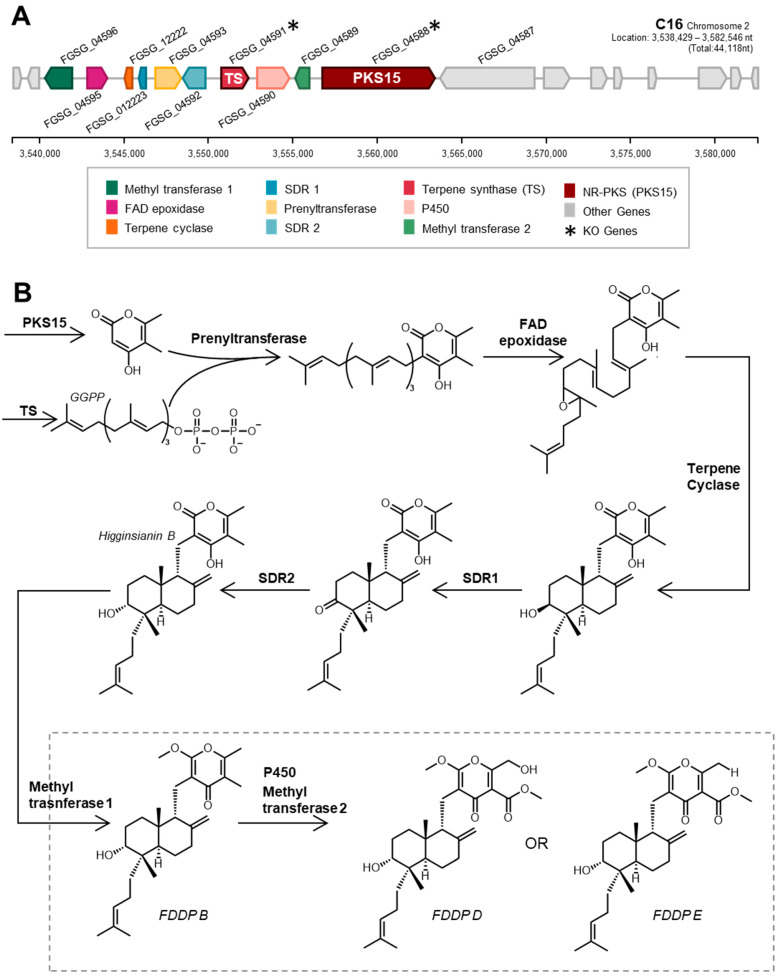
(**A**) Schematic of *F. graminearum* C16 biosynthetic gene cluster and (**B**) biosynthetic steps towards final pathway end products designated as ‘fungal decalin-containing diterpenoid pyrones’ or FDDPs [12].

**Figure 2 jof-09-00695-f002:**
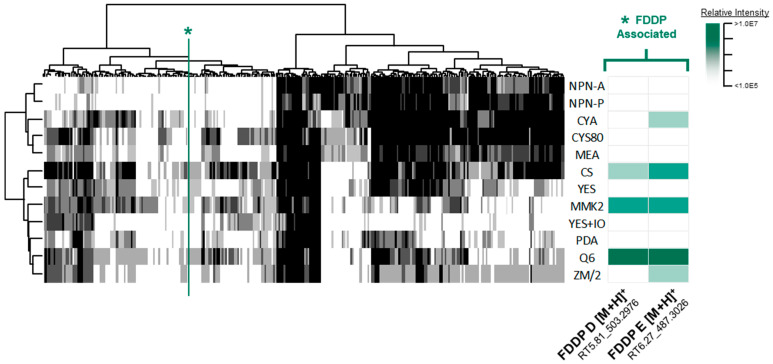
(**left**) Consensus mass feature heatmap of all extraction types comparing media types (columns) to detected mass features (rows) in which the greyscale intensity corresponds to the number of conditions (culture/extraction types) detected for each given medium (black = 4, white = 0). A green asterisk/line is used to indicate FDDP-associated mass features. (**right**) Relative intensity cluster of the FDDP mass features as detected in liquid culture—mycelium extracts (green scale represents mass feature relative intensity).

**Figure 3 jof-09-00695-f003:**
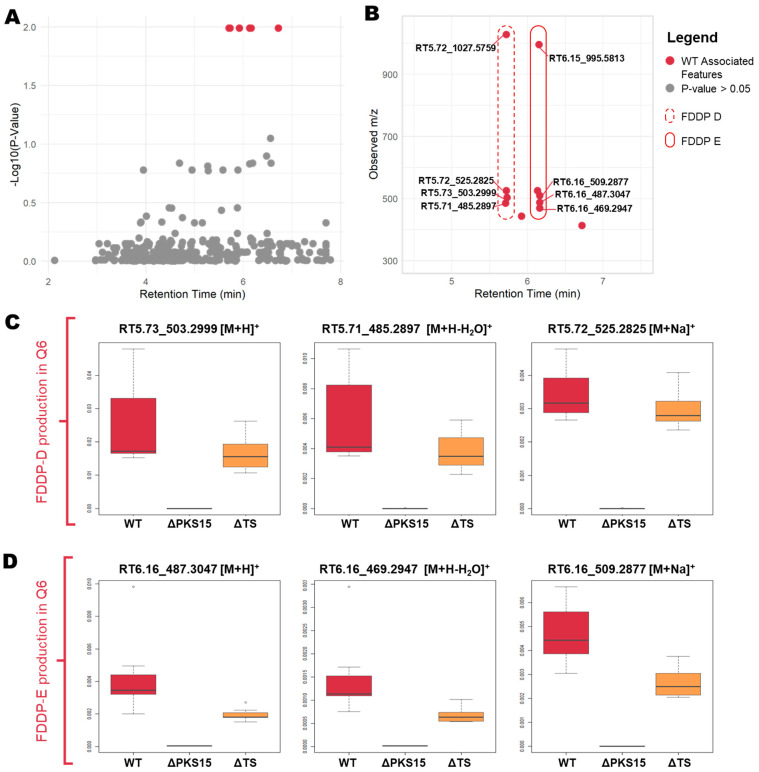
(**A**) All mass features plotted by −log10 (*p*-value) against retention time. Statistically significant variables (*p* < 0.05) are indicated in red. (**B**) Statistically significant variables (*p* < 0.05) plotted by retention time against *m/z* demonstrating associated mass features with putative annotations indicated. (**C**,**D**). Boxplots comparing mass feature intensities of [M + H]^+^, [M + H-H_2_O]^+^, and [M + Na]^+^ pseudomolecular ions between WT, ΔPKS15, and ΔTS strains for production of (**C**) FDDP-D and (**D**) FDDP-E in mycelial extracts from cultures on Q6 medium.

**Figure 4 jof-09-00695-f004:**
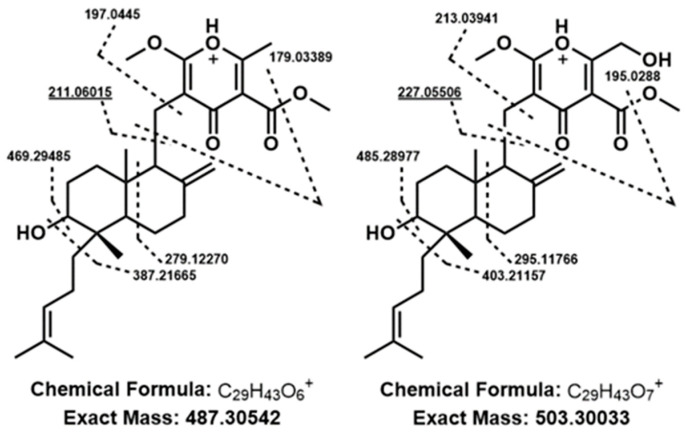
Putative fragmentation patterns of FDDP structures (left to right, FDDP-E, FDDP-D) based on observed MS^2^ mass fragmentation spectra of associated [M + H]^+^ mass features. Underlined *m/z* values were the highest intensity spectra for each feature, correlating to a cleavage between the diterpene and pyrone moieties of the FDDPs.

**Figure 5 jof-09-00695-f005:**
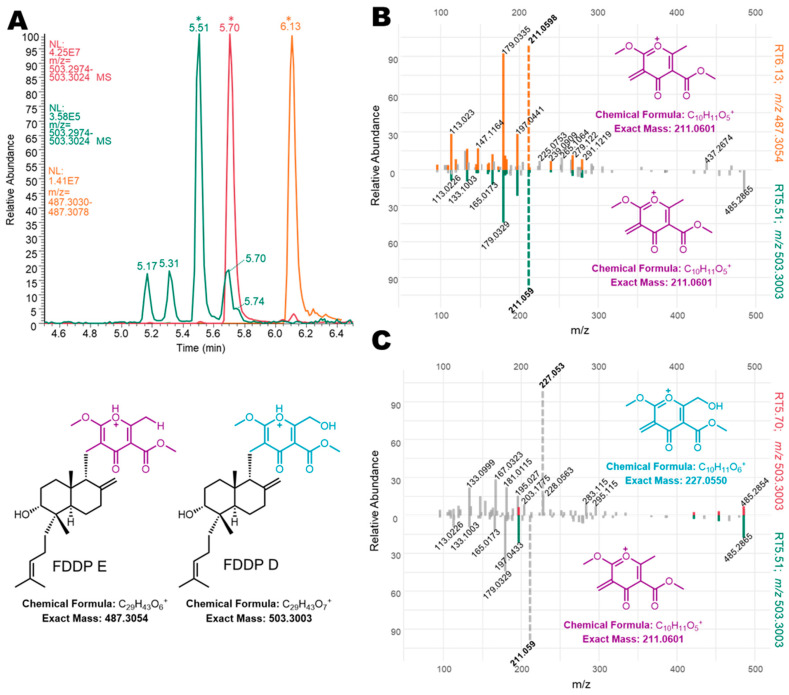
(**A**) Extracted ion chromatograms of mass featured used in MS^2^ mirror plot comparisons with (*) indicating specific retention times. (**B**) Constructed mirror plots comparing FDDP-E (RT6.13; *m/z* 487.3054; top spectrum) from Q6 mycelial extracts to an unknown mass feature detected in wheat extracts (RT5.51; *m/z* 503.3003; bottom spectrum). (**C**) Constructed mirror plots comparing FDDP-D (RT5.70; *m/z* 503.3003; top spectrum) from Q6 mycelial extracts to an unknown mass feature detected in wheat extracts (RT5.51; *m/z* 503.3003; bottom spectrum).

**Figure 6 jof-09-00695-f006:**
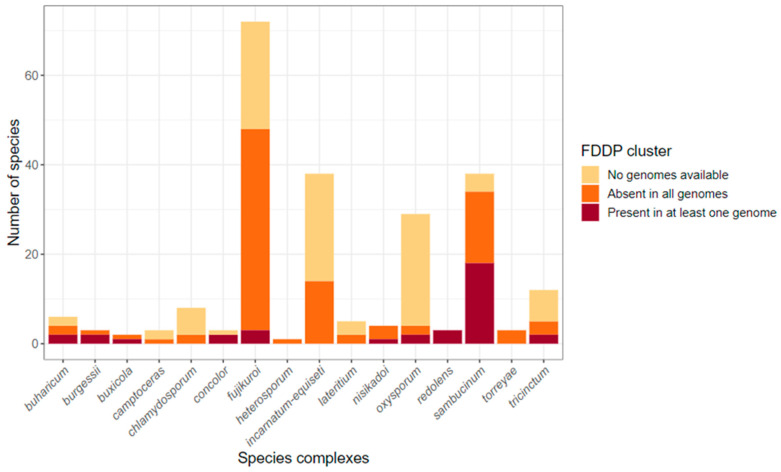
Distribution of C16 biosynthetic gene clusters among *Fusarium* species. Stacked bar plots represent the number of species within each species complex as defined by Crous et al. [26], broken down into whether isolate assemblies showed evidence for a C16 or C16P-like cluster (dark red), did not show evidence (orange), or have not yet had a whole genome sequence published (light tan).

**Figure 7 jof-09-00695-f007:**
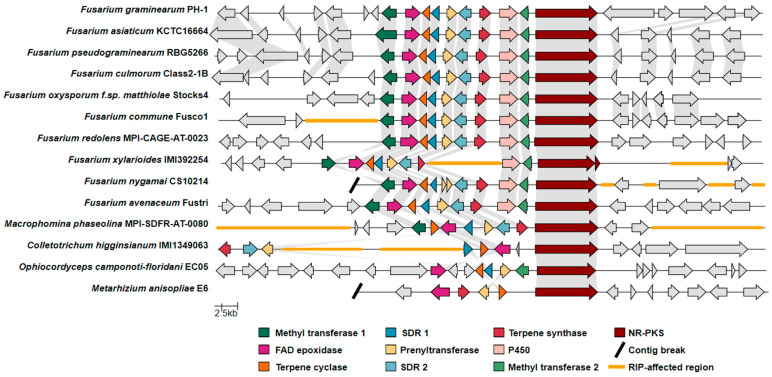
Synteny map of the C16 cluster and adjacent genes, comparing representative strains from relevant *Fusarium* species and other plant-pathogenic fungi. Abbreviations: FAD epoxidase, Flavin adenine-dependent epoxidase; SDR, short-chain dehydrogenase/reductase, P450, cytochrome P450; NR-PKS, nonreducing polyketide synthase; RIP, repeat-induced point mutation.

**Figure 8 jof-09-00695-f008:**
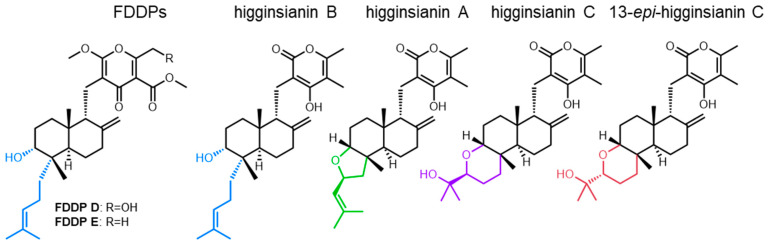
Structural comparisons of FDDP features and higginsianin familial structures with terpene moiety similarities between FDDPs and higginsianin B indicated in Blue and varying structures indicated in green, purple, and red.

## Data Availability

Access to all fungal strains used/generated in this research can be obtained through Agriculture & Agri-Food Canada via consultation with the corresponding author.

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
