# Peer review of "CRISPR-Cas9 Gene Editing and Secondary Metabolite Screening Confirm *Fusarium graminearum* C16 Biosynthetic Gene Cluster Products as Decalin-Containing Diterpenoid Pyrones"

_jof, 2023, doi:10.3390/jof9070695_

Round 1

Reviewer 1 Report

This manuscript describes the functional characterization of PKS15 and adjacent TS (trichodiene synthase) gene in F. graminearum.  This is an excellent manuscript and will be well received.  I have a few minor comments.

L63.  Seems you could tone down references to CRISPR-Cas9.  It’s a tool and does not need to be mentioned here (and in many other sentences).  The point is you sought to determine the product of C16 BGC and you deleted a few genes to do so.

L69.  The biochemical pathway is proposed so please rephrase.  Suggest “ -enzymes likely responsible for…”

L80. Delete fungal. Doesn’t matter where they came from, just a chemical at this point.  Same for L86 and L550.

L93 and L549.  Delete secondary.  The word secondary reflects immediate use by the organism to survive.  Here you are talking about a chemical which says nothing about if its needed for survival.

L95. Replace expression with production.  Expression usually associated with transcription in a gene deletion paper.

L104.  Everything should be thought of as “predicted”, at least until shown by sequencing proteins or by transcript initiation experiments.  Please extend this elsewhere as appropriate.

L 114. What is Littman Oxgall medium?

L122.  Odd wording.  Progression was followed over time, right? Rather than carried out.

L207. knockout seems colloquial.  Would prefer to see “gene deletion mutants” or similar here and elsewhere.   

L236.  Where did you define deltaPKS15 as PKS15 deletion mutant?

L238.  Within? Typo?  Did you mean “with an”?

L348.  confirmed by PCR validation seems redundant.  Do you need validation?

L374.  Should you add expected before production here as don’t yet know what C16 BGC is making?

L 534. Not sure I agree with this statement.  Evidence of RIP indicates a remnant duplicated gene or loci, which can occur anywhere.  i.e. Not restricted to a region of rapid evolution or genomic instability so not evidence there of.

L551. Delete using CRISPR-Cas9.. to end of sentence.

L560 and 561. triple gene deletion mutant rather than knockouts… not really a triple… really a 4-gene mutant even though pks4 and pks13 are near to each other.

L614. Delete homologous.  Or rewrite sentence to clarify.  Yes, TS likely a homolog, but the PKS is definitely not a homolog = different core polyketide.. as are other genes in BGC, which you point out.  Homologous should be save for BGCs that make exactly the same chemical, not just similar chemicals.

I really liked this and next paragraphs, nice work.

Author Response

Dear reviewer,

Thank you for your insightful comments and suggestions regarding our research.  We have carefully considered your comments and made all necessary revisions to address them, as detailed below.

L63.  Seems you could tone down references to CRISPR-Cas9.  It’s a tool and does not need to be mentioned here (and in many other sentences).  The point is you sought to determine the product of C16 BGC and you deleted a few genes to do so.

As this manuscript is being submitted to a Special Issue on Gene-editing Tools in Fungi, an effort was intentionally placed to specifically reference the gene-editing tools that were used in the manuscript.  Therefore we have left in the reference to CRISPR-Cas9 in the introduction section (ln63); however, to take the reviewer’s request into consideration we have removed additional reference to the name CRISPR-Cas9 in other sentences in the manuscript (ln 87, 100, 342, 344).

L69.  The biochemical pathway is proposed so please rephrase.  Suggest “ -enzymes likely responsible for…”

Modified sentence to say “...proposed to be responsible for...”

L80. Delete fungal. Doesn’t matter where they came from, just a chemical at this point.  Same for L86 and L550.

The reference to “fungal” in L80 describes the letter “F” in assigned acronym name for the molecules “FDDP”, as proposed by Tsukada et al. 2020, and explained in this sentence.  We have therefore kept the word, but added quotation marks here to help clarify.  The figure legend line L80 has been similarly altered, as has L550.

L93 and L549.  Delete secondary.  The word secondary reflects immediate use by the organism to survive.  Here you are talking about a chemical which says nothing about if its needed for survival.

The term “secondary metabolite” is widely used in literature when describing small molecules produced by any organism which are not directly associated with normal growth, development, or reproduction of the producing organism. Here “secondary metabolite” is synonymous with the term “natural product”.  It is appropriate to use this term to describe the product of the C16 cluster, because these genes are not associated with primary metabolism (in fact they are missing from many Fusarium species, as described in our work), and furthermore in this study we have removed the ability of the fungus to produce the molecules associated with the C16 cluster by knocking out the core PKS and TS involved.  We observed the lack of FDDP-associated mass features in the resulting mycelia, which was still able to develop normally– therefore by definition these are shown to be ‘secondary metabolites’.

L95. Replace expression with production.  Expression usually associated with transcription in a gene deletion paper.

Replaced ‘expression’ with ‘production’.

L104.  Everything should be thought of as “predicted”, at least until shown by sequencing proteins or by transcript initiation experiments.  Please extend this elsewhere as appropriate.

Added the term ‘predicted’ to line 105.

L 114. What is Littman Oxgall medium?

Littman Oxgall medium is a medium our lab uses to aid in protoplasting, as it has been observed to promote hyphal branching in actively growing mycelia.  Although we have no specific citation for this observation, we have added this info as a note into the supplemental information file, where the specific media formulation for Littman Oxgall medium is detailed.

L122.  Odd wording.  Progression was followed over time, right? Rather than carried out.

Reworded this sentence to read “Protoplast formation was observed periodically over a total incubation time …”.

L207. knockout seems colloquial.  Would prefer to see “gene deletion mutants” or similar here and elsewhere.   

Agreed.  The term gene deletion or transformants were intended to be used throughout the manuscript ….to avoid using knockout ….we obviously missed a few instances however.  Thanks for spotting them….knockout has been replaced with gene deletion throughout the manuscript.

L236.  Where did you define deltaPKS15 as PKS15 deletion mutant?

Altered this line to define the usage of the terms.

L238.  Within? Typo?  Did you mean “with an”?

Typo.  Changed to “with”.

L348.  confirmed by PCR validation seems redundant.  Do you need validation?

Removed the word “validation”.

L374.  Should you add expected before production here as don’t yet know what C16 BGC is making?

Added “expected” before “C16 BGC products”.

L 534. Not sure I agree with this statement.  Evidence of RIP indicates a remnant duplicated gene or loci, which can occur anywhere.  i.e. Not restricted to a region of rapid evolution or genomic instability so not evidence there of.

This line has been removed from the manuscript, as it was unnecessary conjecture.

L551. Delete using CRISPR-Cas9.. to end of sentence.

Done

L560 and 561. triple gene deletion mutant rather than knockouts… not really a triple… really a 4-gene mutant even though pks4 and pks13 are near to each other.

Agreed.  We have corrected the text to refer to a “triple mutant” strain, instead of a “triple knockout”.  This is consistent with how Seidl et al. 2022 described their strains and should be clearer to readers comparing studies.

L614. Delete homologous.  Or rewrite sentence to clarify.  Yes, TS likely a homolog, but the PKS is definitely not a homolog = different core polyketide.. as are other genes in BGC, which you point out.  Homologous should be save for BGCs that make exactly the same chemical, not just similar chemicals.

The PKSes in question are homologous.  The core polyketide moiety is consistent between the higginsianins and FDDPs (please refer to Figure 1 – where higginsianin B is seen as a precursor in the biosynthesis of FDDP D and E).  The FDDP pyrone moiety appears to have different decorations due to the activity of a methyl transferase (MT1) mediating O-methylation of the pyrone, and a P450 which is proposed to oxidize both methyls adorning the pyrone.  Both molecule families can be seen to have identical PKS-derived moieties, and so the two genes should be thought of as homologs.  In our opinion, BGCs should be considered as homologous if they produce the same core molecular scaffold even in the presence of different tailoring reactions – this concept forms the basis of ‘gene cluster families’ which is at the core of pan-genomic BGC analysis via programs such as “BiG-FAM”.  Nevertheless, the literature is less clear on what constitutes a homologous BGC versus BGCs which share homologous genes… therefore we have rephrased our text for clarity and removed reference to ‘homologous BGC” where different tailoring enzymes are present.   

Reviewer 2 Report

In this study, Overy et al. report the correlation of the C16 biosynthetic gene cluster with two decalin-containing, diterpenoid pyrone products by CRISPR-Cas9-beased gene editing and comparative metabolic analyses of the resultant mutants (i.e., ΔPKS15 and ΔTS) and the wild-type Fusarium graminearum strain both in vivo and in planta.

Overall, this manuscript was well written. Efforts for metabolic profiling using 12 different medium formulations to activate the C16 cluster under various culture conditions are particularly impressive. Unfortunately, the two diterpenoid pyrone products (e.g., termed FDDP-D and FDDP-E) are not new and have been structurally characterized couple years ago, and the associated biosynthetic pathway was established clearly based on heterologous expression of the C16 cluster in Aspergillus oryzae (Tsukada, K., et al. Nat. Commun. 2020, 11, 1830). While a distinct MS signal was observed in planta, the structure of the corresponding pyrone product remains to be determined. The authors discussed previous studies showing that the production of related decalin-containing, diterpenoid pyrone products is associated with the invasion process of Fusarium graminearum; however, whether these products serve as virulence factors (or can be combined with other characterized molecules such as deoxynivalenol and fusaoctaxins to afford synergistic effects) relies on further experimental evidence. The above concerns need to be addressed if being considered by the journal.

Other minors:

1.       Page 2, line 44: “Fusarium head blight (FHB)”.

2.       Page 2, lines 60-62: please rephrase/revise the sentence “Proof of the involvement of the F. graminearum C16 BGC products as a virulence factor involved with FHB disease”. Perhaps “Proof of the involvement of the F. graminearum C16 BGC products in the FHB disease as virulence factors”.

3.       Page 10, line 400: “intermediates in the FDDP biosynthetic pathway”.

4.       Page 10, line 405: “compared to WT in the Q6 medium”.

5.       Page 11, line 442: “Extracted ion chromatograms (EICs)”. Same as the others in the main text.

Author Response

Dear reviewer,

Thank you for your review of our work.  We understand your concerns and have addressed them to the best of our abilities below, in addition to all minor text edits.

To address the reviewer comment: “Unfortunately, the two diterpenoid pyrone products (e.g., termed FDDP-D and FDDP-E) are not new and have been structurally characterized couple years ago, and the associated biosynthetic pathway was established clearly based on heterologous expression of the C16 cluster in Aspergillus oryzae (Tsukada, K., et al. Nat. Commun. 2020, 11, 1830). While a distinct MS signal was observed in planta, the structure of the corresponding pyrone product remains to be determined.”

We have discussed the work of Tsukada et al. 2020 at length in this paper, and obtained purified samples of relevant FDDPs from their lab to compare column chromatographic retention time as well as MS2 fragmentation patterns to our experimentally derived data.  Our results clearly match characterized FDDP molecules to our predicted FDDPs from analyses of complex plant extract matrices.  Unfortunately, Fusarium graminearum simply does not produce enough FDDP-associated product in any of the 12 axenic culture conditions we tested, nor in planta, to enable their purification as this reviewer is suggesting.  What makes our work important is that we have added the biological relevance of having detected identical FDDP signals to those described by Tsukada et al. 2020 from plant extracts of infected wheat heads.  It is important to note that Tsukada’s et al. 2020’s characterization of FDDP signals was actually performed using a combination of genes derived from Arthrinium sacchari (core genes) and F. graminearum (tailoring enzymes), expressed in A. oryzae (reference of which has been made in our manuscript).  Therefore, prior to our work it was still a conceptual leap to assume that FDDPs characterized by Tsukada et al. 2020 are produced by F. graminearum.  We have not merely characterized “a distinct MS signal” in planta, as suggested in your review – we have characterized MS signals that are identical to the purified molecules characterized by Tsukada et al.  It is well beyond the scope of this work to scale fermentations to the extent that we would be able to purify all FDDPs discussed from any of the conditions tested. We agree that this would strengthen the research, however it is not feasible at this time, and does not detract from the relevance of our manuscript as it is written. Based on our MS/MS data to confirm the pyrone portion of the unknown and comparison of the various terpenoid moieties observed for the higginsianins, in particular higginsianins C and epi-C, we have a hypothesis as to the structure of the unknown metabolite from the in planta extracts; however, with a full structure elucidation effort by NMR, we chose not include this information in the manuscript

The authors discussed previous studies showing that the production of related decalin-containing, diterpenoid pyrone products is associated with the invasion process of Fusarium graminearum; however, whether these products serve as virulence factors (or can be combined with other characterized molecules such as deoxynivalenol and fusaoctaxins to afford synergistic effects) relies on further experimental evidence. The above concerns need to be addressed if being considered by the journal.

Our discussion of FDDPs as virulence factors with the potential to act synergistically with other molecules is just that – a discussion.  We clearly describe in the discussion section how our plant disease phenotype results are preliminary, and are ‘suggestive’ of a virulence factor role for these secondary metabolites.  At no point have we claimed this to be proven by our work.  This part of the discussion is important to our paper because it sets the stage for future research to be undertaken, investigating the potential virulence effects of FDDPs, by themselves or in combination with other putative virulence factors. This is not a shortcoming of our research paper, it is a framing of a hypothesis built from all contemporary knowledge on the C16 cluster’s expression, in combination with our preliminary observations of disease phenotype shifts brought on by deletion of select C16 genes.  If the reviewer believes we have overstepped in our discussion, they are welcome to be more specific as to what they would like to see changed.

Other minors:

  1. Page 2, line 44: “Fusarium head blight (FHB)”.

Corrected.

  1. Page 2, lines 60-62: please rephrase/revise the sentence “Proof of the involvement of the F. graminearum C16 BGC products as a virulence factor involved with FHB disease”. Perhaps “Proof of the involvement of the F. graminearum C16 BGC products in the FHB disease as virulence factors”.

The line has been revised to read: “Proof of the involvement of the F. graminearum C16 BGC products as FHB-associated virulence factors remains to be conclusively established.”

  1. Page 10, line 400: “intermediates in the FDDP biosynthetic pathway”.

Revised.

  1. Page 10, line 405: “compared to WT in the Q6 medium”.

Revised.

  1. Page 11, line 442: “Extracted ion chromatograms (EICs)”. Same as the others in the main text.

As requested, all references to “XIC” have been changed to “EIC”.

Round 2

Reviewer 1 Report

No additional comments to authors.  And yes, I see/agree that they are homologs.

Reviewer 2 Report

The authors tried their best to address the concerns from the reviewers and the manuscript was improved as required.